# Use of Antimicrobials for Bloodstream Infections in the Intensive Care Unit, a Clinically Oriented Review

**DOI:** 10.3390/antibiotics11030362

**Published:** 2022-03-08

**Authors:** Alexis Tabah, Jeffrey Lipman, François Barbier, Niccolò Buetti, Jean-François Timsit

**Affiliations:** 1Intensive Care Unit, Redcliffe Hospital, Metro North Hospital and Health Services, Redcliffe, QLD 4020, Australia; 2School of Clinical Sciences, Queensland University of Technology, Brisbane, QLD 4000, Australia; 3Antimicrobial Optimisation Group, UQ Centre for Clinical Research, The University of Queensland, Brisbane, QLD 4029, Australia; j.lipman@uq.edu.au; 4Jamieson Trauma Institute and Intensive Care Services, Royal Brisbane and Women’s Hospital, Metro North Hospital and Health Services, Brisbane, QLD 4029, Australia; 5Division of Anaesthesiology Critical Care Emergency and Pain Medicine, Nîmes University Hospital, University of Montpellier, 30029 Nîmes, France; 6Medical Intensive Care Unit, CHR Orléans, 45100 Orléans, France; barbierfrancois.chro@gmail.com; 7IAME, INSERM, Université de Paris, 75018 Paris, France; niccolo.buetti@hcuge.ch (N.B.); jean-francois.timsit@aphp.fr (J.-F.T.); 8Infection Control Program and WHO Collaborating Centre on Patient Safety, Geneva University Hospitals and Faculty of Medicine, Rue Gabrielle-Perret-Gentil 4, 1205 Geneva, Switzerland; 9APHP Medical and Infectious Diseases Intensive Care Unit (MI), Bichat-Claude Bernard Hospital, 75018 Paris, France

**Keywords:** bloodstream infection, bacteraemia, sepsis, septic shock, empirical, probabilistic antibiotics, source control, de-escalation, ICU, intensive care

## Abstract

Bloodstream infections (BSIs) in critically ill patients are associated with significant mortality. For patients with septic shock, antibiotics should be administered within the hour. Probabilistic treatment should be targeted to the most likely pathogens, considering the source and risk factors for bacterial resistance including local epidemiology. Source control is a critical component of the management. Sending blood cultures (BCs) and other specimens before antibiotic administration, without delaying them, is key to microbiological diagnosis and subsequent opportunities for antimicrobial stewardship. Molecular rapid diagnostic testing may provide faster identification of pathogens and specific resistance patterns from the initial positive BC. Results allow for antibiotic optimisation, targeting the causative pathogen with escalation or de-escalation as required. Through this clinically oriented narrative review, we provide expert commentary for empirical and targeted antibiotic choice, including a review of the evidence and recommendations for the treatments of extended-spectrum β-lactamase-producing, AmpC-hyperproducing and carbapenem-resistant Enterobacterales; carbapenem-resistant *Acinetobacter baumannii;* and *Staphylococcus aureus*. In order to improve clinical outcomes, dosing recommendations and pharmacokinetics/pharmacodynamics specific to ICU patients must be followed, alongside therapeutic drug monitoring.

## 1. Introduction

A bloodstream infection (BSI) is defined as the microbial invasion of the blood stream. In clinical practice, this refers to a positive blood culture (BC) from a patient with clinical signs of infection [1]. Bloodstream infections can be categorised in a range of meaningful ways: According to the origin of the infection, either community-acquired (CA-BSI), hospital-acquired (HA-BSI) or intensive care unit (ICU)–acquired (ICU-BSI).Either secondary to a source of infection or primary, when there is no identified source [2].Complicated or uncomplicated, which was recently defined as a having definite source (among urinary, catheter, intra-abdominal, pneumonia, skin or soft tissues), and effective source control, in a non-immunocompromised patient, and with clinical improvement after 72 h of antimicrobial therapy (at least defervescence and haemodynamic stability) [3].By clinical severity, which is the absence or presence of organ failures and the need for organ supportive therapy in the ICU.

Critically ill patients are often debilitated and suffer from immune paresis caused by their initial reason for ICU admission [4]. Secondary infections are especially common in patients with higher severity of disease [5]. For ICU patients, BSIs are associated with significant mortality, ranging from 35% to more than 60% [6,7,8]. In a cohort study of 10,734 ICU patients with an ICU length of stay (LOS) of more than 3 days, 571 (5.3%) developed ICU-BSIs. In a multivariable COX model analysis, ICU-BSIs were independently associated with increased mortality [8]. 

This clinically oriented narrative review focusses on the antimicrobial management of BSIs, whose clinical severity requires ICU admission, or such infections that have been acquired in the ICU. We will review the importance of microbiology specimens, the timing and choice of the empirical antimicrobial therapy, the role of spectrum and dose optimisation, the importance for source control and, finally, strategies for stopping antimicrobials (Figure 1).

## 2. Antimicrobial Therapy

### 2.1. Empirical Antimicrobial Therapy 

#### 2.1.1. The Importance of Getting It Right from the Start

For ICU patients with sepsis or septic shock, it is recommended to administer antimicrobials immediately, ideally within one hour of recognition [9]. This is supported by observational data. Kumar and colleagues described in 2006 a 12% increase in crude mortality for each hour of delay to administer antimicrobials from the onset of hypotension and septic shock [10]. The above-mentioned study by Adrie and colleagues shows a 30% increase in mortality when no adequate treatment is given in the first 24 h [8]. In the evaluation of a multifaceted intervention to decrease sepsis mortality in a group of 40 German hospitals, Bloos and colleagues report an increase in the risk of death of patients with sepsis or septic shock of 2% for each hour of delay of antimicrobial therapy and 1% for each hour of delay in source control [11]. However, not all research on time to antibiotics has been so positive [12]. Hranjec and colleagues investigated the issue with a before and after study in surgical ICU patients with sepsis but without shock [13]. They compared an aggressive approach where antibiotics were started as soon as sepsis was recognised to a conservative approach where they were started only if the infection was confirmed by positive microbiology. In the conservative period, immediate antibiotic therapy was recommended for patients with shock. The aggressive approach was associated with a lower time from fever and BC to start of treatment. The conservative approach was associated with more initially appropriate therapy, a shorter duration of antibiotics and lower mortality. This manuscript demonstrates the difficulty intensivists face daily in trying to differentiate infection from inflammation in the ICU patient population. It is conceivable that several patients were without infection and, therefore, did not require antibiotics. Delaying antibiotics to investigate the cause of “sepsis” may have multiple benefits for patients with low severity. It may improve outcomes through the diagnosis and management of non-infectious causes of inflammation and organ failures plus avoid harm from antibiotic overuse. Further, it will help in obtaining a diagnosis for a proportion of infections that would otherwise been labelled as “culture negative” or “from unknown source”. Definitive clinical and microbiological diagnosis of an infection facilitates the provision of a targeted treatment and improves outcomes. While controversy remains and these data present all the biases inherent to observational studies, they highlight how important it is that patients with BSIs receive early appropriate antimicrobial therapy. 

#### 2.1.2. Broad-Spectrum Antibiotics and Combination Therapy?

The empirical regimen should be broad enough to maximise the likelihood of adequacy, especially in patients with septic shock. This may, however, lead to an unnecessary overuse of broad-spectrum antimicrobials and associated harms, including the promotion of antimicrobial resistance [14].

When the source is known, antibiotics should be targeted at the most common pathogens for the source as detailed in Table 1. Molecule choice takes into account risk factors for multidrug-resistant (MDR) or specific pathogens for the patient, according to their history and setting as shown in Table 2. For hospital-acquired infections, knowledge of colonisation from previous clinical or surveillance cultures is a valuable tool to optimise this choice [15,16].

Combination therapy can provide very broad empirical coverage for different classes of pathogens by adding anti-MRSA and antifungal agents or molecules targeted at MDR Gram-negative bacteria (GNB). These should be used with parsimony, in patients with significant risk factors, and only as part of the empirical regimen with a plan to subsequently de-escalate all drugs that are not required [17,18].

#### 2.1.3. The Importance of Sending Blood Cultures before Starting Antimicrobials

The empirical antibiotic choice is made while differential diagnosis is still underway, including uncertainty on the pathogen. Microbiology results will be required to judge of the presence of an infection and to optimise antimicrobial therapy by targeting the causal pathogen(s) or to stop antibiotics if there is no infection. 

Sending specimens before starting antimicrobials (without delaying the treatment) is key to avoiding false-negative results. Sheer and colleagues analysed the factors associated with BC positivity in a single centre cohort study of 599 patients with severe sepsis or septic shock who had at least two BC sets taken [19]. Patients with cultures sampled before antibiotics had a 50.6% positivity rate, almost double the 27.7% for those who had received antibiotics before. They showed that antibiotic therapy prior to BC sampling was an independent factor for BC negativity. In this cohort, 35 patients had cultures sampled both before and after antibiotics. The positivity rate was 57.1% (20/35) before antibiotics. After antibiotics, positivity decreased to 25.7% as 9 of those 20 patients still had positive cultures. This represents a loss of pathogen detection of 30.0% and highlights the importance of sending cultures prior to starting antibiotics. 

When antibiotics are indicated, and the patient has septic shock, taking cultures must not delay the initiation of antimicrobials beyond a reasonable delay of 15 to 45 min [9]. Importantly, clinicians should not wait for culture results to start the treatment. When an ICU patient develops new signs of sepsis, cultures should be sent from the likely source(s) of infection, from the blood and most often also from urine and sputum.

Sampling quality is very important. We recommend at least two sets of aerobic and anaerobic BCs, from two different sites, inoculating a sufficient amount of blood per bottle [2], usually, 8–10 mL per bottle. It is, however, good practice to check manufacturers’ recommendations. Blood should be sampled peripherally following rigorous skin disinfection, and an aseptic non-touch technique for drawing the blood and inoculating the bottles is key to decreasing false-positive results from BC contamination with commensal micro-organisms [20]. 

#### 2.1.4. The Advent of Molecular Rapid Diagnostic Testing

The rapidly expanding field of molecular rapid diagnostic testing (mRDT) provides a range of diagnostic tools for the faster identification of pathogens and specific resistance patterns from the initial positive BC [21]. A laboratory requires up to 1–2 days to identify the micro-organism from a positive BC and another 1–2 days to provide the antibiogram [22]. Accurate bacterial species identification is available in the matter of hours with techniques such as matrix-assisted laser desorption/ionisation–time of flight (MALDI-TOF) mass spectrometry [23]. Integrated solutions such as the Accelerate Pheno system automate both the identification and AST, providing accurate results in 90 min and 7 h, respectively. In a multicentre study, comparing with conventional BC processing, it accurately identified 14 common bacterial pathogens and 2 *Candida* sp. with sensitivities ranging from 94.6% to 100% [24]. The performance of AST results for methicillin-resistant *Staphylococcus aureus* (MRSA) and *Staphylococcus* sp. had an agreement of 97% with conventional processing. For GNB, the agreement on a panel of 15 antimicrobials was 94%, making this system suitable for prime clinical use [24]. 

Colorimetric assays are relatively inexpensive and extremely accurate benchtop solutions to detect extended-spectrum beta-lactamase-producing (ESBL-Es) or carbapenemase-producing Enterobacterales (CPEs) [21,25]. The newest kits such as the NitroSpeed-Carba NP can identify the presence and production of carbapenemase by GNB with a sensitivity of 100% and a specificity of 97%. It detects the type of carbapenemase with sensitivities ranging from 97% to 100%, even in cases with a very low level of carbapenemase activity [26]. These may allow for the urgent escalation of antibiotics, gaining several hours to days when compared with waiting for an antibiogram. When used within an antimicrobial stewardship (AMS) program, they may help to avoid the over prescription of the newer β-lactam–β-lactamase inhibitors (BL/BLIs) in the empirical regimen. Their use for ADE can (and should) be done, with caution as clinical evidence is only emerging [18].

### 2.2. What to Do with Culture Results

Patients with a suspected and then confirmed BSIs need to have microbiology results reviewed at least daily. The antibiotic treatment must be targeted to the pathogen in terms of molecule activity, with an adequate penetration at the source and sufficient dosing, as early as possible, and for the whole duration of the treatment, without exceeding the required duration. Effective communication with the microbiology laboratory is crucial. In our practice, we check for results during the morning and afternoon rounds, and the laboratory will call us almost immediately when they have a positive BC or any significant result. Antimicrobial stewardship programs and scheduled infectious diseases rounds help to ensure that no opportunities to optimise the treatment are missed.

The initial communication by the laboratory of a positive BC and Gram stain results may be the time when antibiotics are started or escalated. Identification of the pathogen comes a few hours to a day later and may include information on mechanisms of resistance depending on laboratory technique availability. Lastly, we will receive an antibiogram and final confirmation of the identified pathogen. At each step, we ensure the causative pathogen is covered by the administered treatment. With the final microbiology results, we make a definitive adjustment to the antibiotic regiment, including a decision on the duration of therapy.

Antimicrobial de-escalation (ADE) consists in either (i) replacing a broad-spectrum antimicrobial with an agent of a narrower clinical spectrum or a presumed lower ecological impact or (ii) stopping a component of an antimicrobial combination [18]. It is an important tool to reduce the exposure to broad-spectrum antibiotics and prevent the emergence of antimicrobial resistance. Antimicrobial de-escalation has demonstrated patient-level safety, with a meta-analysis suggesting improved outcomes in patients who received ADE [27]. Bloodstream infections are very specific as the causing pathogen is known with certainty, and this makes them perfect targets for ADE. Some sources, such as peritonitis or deep-seated abscesses may be polymicrobial, with sometimes the indication to maintain broader cover for some suspected—but not grown—pathogens. In nearly all other situations, we can safely select the molecules that provide the most adequate treatment for the pathogen causing the BSIs at the source, while having the lowest ecological impact. Importantly, outside specific extensively drug-resistant (XDR) pathogens, there is no benefit to continuing combination therapy for GNB infections [28].

#### 2.2.1. Specific Pathogens

While ADE and narrow-spectrum antibiotics can be easily recommended for susceptible micro-organisms, globally increasing antimicrobial resistance (AMR) has significantly complicated antibiotic management as detailed in the examples below.

##### Extended-Spectrum β-Lactamase-Producing Enterobacterales

ESBL-producing Enterobacterales (ESBL-Es), and their surrogate, Enterobacterales resistant to third-generation cephalosporins should be treated with a carbapenem [29,30]. Carbapenem sparing in this context has been extensively investigated and was initially supported by observational studies [31]. The MERINO trial randomised 391 patients with a BSIs due to ceftriaxone-resistant *Escherichia coli* or *Klebsiella pneumoniae* to piperacillin–tazobactam or meropenem [32]. Mortality was 12.3% for piperacillin–tazobactam compared with 3.7% for meropenem, rejecting non-inferiority and not supporting the use of piperacillin–tazobactam in severe infections due to ESBL-Es. Alternatives for cases where a carbapenem cannot be used include fluoroquinolones and trimethoprim-sulphamethoxazole. Those are especially interesting for BSIs with a urinary source as they concentrate in the urine [30]. While ceftolozane–tazobactam and ceftazidime–avibactam (CAZ-AVI) are potential alternatives, their use should be restricted as reserve antibiotics for those pathogens that cannot be treated otherwise. 

##### Inducible AmpC-Producing Enterobacterales 

Enterobacterales including *Enterobacter cloacae*, *Klebsiella aerogenes* (ex. *Enterobacter aerogenes*) and *Citrobacter freundii* are the main pathogens of concern that carry a chromosomal inducible AmpC β-lactamase [33]. These are problematic because they initially show susceptibility to ceftriaxone. However, exposure to ceftriaxone and other β-lactams such as piperacillin–tazobactam or imipenem will induce a sufficient increase in the production of AmpC to cause resistance to ceftriaxone, leading to treatment failure [33,34]. These enzymes effectively hydrolyse ceftriaxone and ceftazidime. Tazobactam has weak efficacy against AmpC β-lactamases, and observational studies were equivocal [35]. The MERINO-2 pilot trial randomised patients with AmpC BSIs to piperacillin–tazobactam or meropenem. There was numerically higher mortality and clinical and microbiological failure with piperacillin–tazobactam but more relapses with meropenem. Pending further data, we should avoid using piperacillin–tazobactam in patients with severe infections due to pathogens with inducible AmpC [36,37]. Cefepime is a good treatment choice as it is a weak inducer, and it is relatively stable against AmpC β-lactamases. Caution is warranted in pathogens with a MIC ≥ 4 µg/mL for cefepime as they may harbour an ESBL, making them prone to treatment failure. All carbapenems are stable and recommended for the treatment of AmpC-hyperproducing Enterobacterales. New β-lactamase inhibitors (BLIs), such as avibactam, are very effective, but their use should be restricted to pathogens that do not have other treatment options [33]. For pathogens that are susceptible, fluoroquinolones and trimethoprim-sulphamethoxazole can be considered as alternatives [36]. 

##### Carbapenem-Resistant Enterobacterales

Carbapenem-resistant Enterobacterales (CREs) are defined by resistance to at least one carbapenem [38]. This can be either due to the production of a carbapenemase, such as *Klebsiella pneumoniae* carbapenemases (KPCs), oxacillinase (e.g., OXA-48), and metallo-β-lactamases (MBLs) (e.g., New Delhi metallo-β-lactamases), or a combination of other mechanisms, such as a mutation in porin genes that limit the entry of the antibiotic into the bacteria associated with upregulated production of other β-lactamases [39]. 

Recently, combinations of older β-lactams with a new BLI and a novel cephalosporin have been marketed specifically for the management of CREs. Avibactam, in CAZ-AVI is targeted to the inhibition of KPCs and OXA-48 carbapenemases. It is inactive against MBLs. Ceftazidime–avibactam was shown to be effective in a cohort study of 137 patients with infections caused by a CRE. There was an inverse probability of treatment weighting (IPTW)–adjusted probability of a better outcome of 64% with CAZ-AVI when compared with colistin [40]. While no randomised controlled trial (RCT) is available to date, these results are concordant with other studies comparing CAZ-AVI with other antibiotics [41,42].

Meropenem–vaborbactam is targeted at KPCs but is inactive against OXA-48 and MBLs. It was investigated in a 77-patient phase-3 RCT against the best available treatment (BAT) [43]. Forty-four patients had confirmed CRE infections. In this subpopulation, meropenem-vaborbactam was associated with improved cure rates (59.4% vs. 26.7%, *p* = 0.002) and a numerically but not statistically lower mortality (15.6% vs. 33.3%, *p* = 0.2). 

There is less evidence for cefiderocol, a siderophore cephalosporin active in vitro against all CPEs including MBLs. The CREDIBLE-CR RCT included 118 patients with a CR-GNB at baseline (46% *A. baumannii*, 33% *K. pneumoniae* and 19% *P. aeruginosa*) compared cefiderocol and BAT for CR-GNB [44]. Mortality was higher in the cefiderocol arm (24.8% vs. 18.4%). A subgroup analysis showed that higher mortality was found in patients with carbapenem-resistant *Acinetobacter baumannii* (CRAB) but not in those with CREs [30]. When comparing cefiderocol with BAT, clinical cure was 66% vs. 45% in the CRE subgroup and 75% vs. 29% in the MBL subgroup. Aztreonam–avibactam is a very promising combination with potent activity against multiple carbapenemases including MBLs [45]. It is unfortunately not yet available for broad clinical use. Some MBLs that are resistant to cefiderocol, CAZ-AVI and other BL/BLIs remain susceptible in vitro to the combination of ceftazidime–avibactam–aztreonam. This treatment was independently associated with lower 30-day morality in an observational study of 102 patients with MBL-producing CRE BSIs and, with cefiderocol, may be, one of the only available treatment options for MBL producers [46,47].

Given the specific activity of each of those antimicrobials, effective AMS and use of phenotypic tests to determine the presence of each resistance mechanism are important to manage CRE BSIs in the ICU.

For CRE strains that are susceptible to BL/BLIs, there is no indication to add a second antibiotic as part of combination therapy, and if one was started, we suggest ADE [30]. A recent propensity-matched cohort study of 577 patients with KPC-producing *K. pneumoniae* (KPC-Kp) treated with CAZ-AVI combination therapy did not show benefit versus CAZ-AVI monotherapy [41]. This contrasts with studies published before the advent of the new generation of BL/BLIs. The INCREMENT cohort showed in the high-mortality risk strata of patients with CRE-BSIs an independent association between combination therapy and a lower risk of death [48]. When antibiotics such as polymyxin and tigecycline are used as pivotal antibiotics, combination therapy remains advised [30,48]. 

##### Carbapenem-Resistant *Acinetobacter*
*baumannii*

Carbapenem-resistant *Acinetobacter baumannii*, and other *Acinetobacter* sp. resistant to carbapenems have very limited treatment options and subsequently high risks of treatment failure and mortality [49]. This is due to the common co-existence of multiple mechanisms conferring combined resistance to most or all antibiotic classes [36,49]. Further, the efficacy of novel BL/BLI combinations is disappointing. Vaborbactam does not restore the activity of meropenem against CRAB. Relebactam does not improve the activity of imipenem. Noting that CAZ-AVI is not indicated for CRAB, we refer to a study of 71 U.S. hospitals in 2012–13 finding that up to 73.6% of CRAB from ICU isolates were resistant to CAZ-AVI [50]. 

This pathogen remains one of the few indications in which it may be indicated to continue combination therapy for the duration of the treatment or at least until clinical improvement [36]. Combinations should include in vitro active drugs, where available. Given the paucity of treatment options, multiple combinations have been tested. A multicentre RCT compared colistin alone or combined with meropenem (both administered at high doses) and found no difference in terms of clinical failure or 28-day mortality [51], not supporting the addition of meropenem to colistin for CRAB. Sulbactam has specific intrinsic antibiotic activity against *Acinetobacter* sp. For susceptible isolates, ampicillin–sulbactam is the preferred choice as the pivotal antibiotic of a combination regimen [52]. These strains are, however, becoming rare, and polymyxins are often one of the few available options. Polymyxin B is recommended for systemic infections because of better pharmacokinetic (PK) characteristics and less nephrotoxicity than colistin methane sulphonate (CMS), which is preferred for urinary sources [36]. Dosing recommendations from the latest guidelines should be followed given their narrow therapeutic index [53]. Tigecycline, if used, should be part of a combination as its clinical efficacy remains debated and its PK profile is unfavourable, especially in the blood and lung tissues [49]. High dosing schemes (200 mg loading followed by 100 mg 12 h) must be employed with caution, and fibrinogen levels must be followed due to time-dependent associated risk of coagulopathy, and dose-dependent gastro-intestinal side effects [54,55]. The adjunction of sulbactam as part of combination therapy for severe infections with strains that are non-susceptible to ampicillin–sulbactam might be considered due to its capacity to saturate altered penicillin-binding protein targets [56].

##### *Staphylococcus aureus* and MRSA

*Staphylococcus aureus* has a propensity for causing HA-BSIs as a complication of medical and surgical procedures or intra-vascular catheters. It often leads to complicated infections, seeding into abscesses, osteoarticular infections and endocarditis, thus, often requiring an extended duration of antibiotics. *S. aureus* can be susceptible to methicillin and many other β-lactam antibiotics (MSSA) or resistant to almost the whole class for MRSA. Newer cephalosporins such as ceftaroline and ceftobiprole have specific anti-MRSA activity. The therapeutic standard for MSSA is a narrow-spectrum anti-staphylococcal β-lactam such as flucloxacillin, oxacillin or a first-generation cephalosporin such as cephazolin [57]. Monotherapy with vancomycin is inferior to β-lactams [58]. In high-prevalence settings, probabilistic treatment should include optimal cover for both MSSA and MRSA. This can be achieved with a combination of flucloxacillin and vancomycin, ceftaroline or daptomycin. There must be a plan for ADE and only the targeted molecule should be retained once the antibiogram is available. 

Vancomycin is the first-line antibiotic for MRSA BSIs [59]. Daptomycin is proposed as a first-line alternative to vancomycin by the Infectious Diseases Society of America (IDSA) guidelines [60]. Linezolid is not recommended as it failed to show non-inferiority to vancomycin in an RCT of MRSA catheter-related BSIs (CR-BSIs) [61]. It may be an option for oral step-down when extended treatments are indicated [62]. Daptomycin was associated with significantly lower rates of clinical failure and 30-day mortality in a propensity-matched cohort of 262 MRSA BSIs [63]. Further, it causes less AKI than vancomycin [64]. However, daptomycin is inactivated by pulmonary surfactant, limiting its indications. The emergence of resistance to daptomycin during treatment may lead to failure and warrants caution. Combination of daptomycin plus ceftaroline as rescue therapy for refractory MRSA BSIs has been reported [65]. Adjunctive rifampicin has long been advocated in MRSA infections to reduce the risk of treatment failure and recurrences but was recently shown to be of no benefit in a large multicentre RCT [66]. 

There is a strong relationship between the duration of bacteraemia and subsequent risk of death [67]. *Staphylococcus aureus* requires extended treatment durations as discussed below. Persisting BSIs may be secondary to endocarditis, and all *S. aureus* BSIs should have a cardiac echocardiography. Transoesophageal echocardiography can only be avoided in cases with specific protective factors [68]. 

## 3. Do Not Forget Source Control

Source control is equally essential with antibiotics in the treatment of BSIs. Surgical or percutaneous management of any abscess, deep-space infection or infected material such as intra-vascular catheters is a matter of urgency. In the EUROBACT International Cohort Study of ICU patients with HA-BSIs, not achieving source control was an independent predictor of day-28 mortality [6]. In a large multicentre observational study of *S. aureus* BSIs, delayed source control was associated with persistent BSIs [57]. In a cohort study of patients with peritonitis and septic shock, delay to surgical source control was an independent predictor of mortality [58]. In unstable patients, the multidisciplinary discussion with the surgical team revolves around timing and choice of the intervention. Damage-control surgery is often indicated. The essential parts of the operation are urgently performed, and a reoperation is planned after clinical stabilisation, 24–48 h later, for a second look and, where possible, anatomical reconstruction [59]. We must emphasise the need to send specimens from the foci of infection at each intervention.

## 4. Optimisation and Dosing Strategies

Sufficient antibiotic concentrations at the site of infection are required for optimal clinical outcomes. The initial and/or loading dose should be given in full, not adjusting for renal impairment [60]. Sepsis alters the PK properties of hydrophilic molecules (β-lactams, glycopeptides and aminoglycosides). They have an increased volume of distribution (Vd) leading to a lower-than-expected maximum serum concentration during a dosing interval (Cmax) [61]. Additionally, augmented renal clearance (ARC), the increase in renal blood flow that often arises in septic shock, leads to the augmented elimination of renally excreted antibiotics. This causes a lower-than-expected trough serum concentration (Cmin). Conversely, renal or hepatic dysfunction may alter the metabolism and elimination of antibiotics, leading to increased concentrations and potential toxicity. Renal replacement therapy (RRT) and extracorporeal membrane oxygenation (ECMO) will also affect the PK of antibiotics, often in unpredictable ways, and require additional monitoring [62].

Further, different PD targets need to be taken in account to ensure maximum bacterial killing and decrease the emergence of resistance. As shown in Figure 2, some antibiotics are concentration dependent (aminoglycosides) and require a high peak concentration obtained with a single daily loading dose. β-Lactams are both concentration and time dependent, requiring sufficient time with a free unbound drug minimum concentration (*f*Cmin) above the MIC for targeted bacteria (*f*Cmin/MIC) [63]. Others such as fluoroquinolones and vancomycin are both time and concentration dependent, and adjustment is based on the ratio of the area under the concentration–time curve from 0 to 24 h to minimum inhibitory concentration (AUC0–24/MIC) [64]. Based on those PK/PD considerations and a meta-analysis of three RCTs suggesting improved short-term mortality [65], it is now suggested to use a prolonged infusion for β-lactams following an initial bolus dose [9]. Further, initial dosing should follow recommendations tailored for critically ill patients (when they are available) rather than following package inserts (Table 3). 

Drug concentrations are usually measured for two reasons, to prevent (or explain) toxicity and to measure for efficacy. Aminoglycosides and glycopeptides have significant side effects at higher concentrations, and hence, measurement facilities are commonly available. More recently, efficacy targets have been set for these antibiotics. Beta-lactams have a high therapeutic ratio with some, but limited, toxicity. Recently, measurements of these compounds have become more relevant in view of underdosing, i.e., therapeutic drug monitoring (TDM) used for the efficacy of these agents [66]. Whilst beta-lactam concentration targets were obtained initially from animal data, there is still debate on what target beta-lactam levels should be used for clinical efficacy [64].

## 5. When and How to Stop Therapy

Minimising the duration of exposure to antimicrobials is important to optimise patient outcomes. A recent umbrella review established how each additional day of therapy is associated with measurable harm [67]. This includes a 4% daily increase in the odds of an adverse drug reaction (OR 1.04, 95% CI 1.02–1.07) and a 3% increase in the odds of antimicrobial resistance (OR 1.03, 95% CI 0.98–1.07). 

Since 2019, three multicentre RCTs with concordant results have established that a 7-day treatment was not inferior to a 10- or 14-day treatment for patients with an uncomplicated GNB BSIs [68,69,70]. We highlight that all patients included in all three RCTs were immunocompetent, afebrile after 3 days of therapy and without uncontrolled infectious sources or prosthetic devices. For ICU patients with BSIs, the duration of therapy should be individualised based on clinical response. A rapid decrease of biomarkers such as PCT or CRP might be interesting to reduce the duration of therapy [68,71,72]. For uncomplicated GNB BSIs, it is not necessary to send repeat BC to ensure bacterial clearance [3]. Otherwise, at least one set of BCs sent at day 2–4 is required. For *S. aureus*, multiple negative BCs may be required to ensure BSIs clearance [73].

Persisting bacteraemia is defined as 2 days or more with positive BC despite active antibiotics [74]. For those cases, after ensuring the pathogen is not resistant to the administered antibiotic, we need to repeat clinical examination and investigations (e.g., CT scanner) looking for a source that had been missed such as a deep-seated abscess. A cardiac echography may be necessary to exclude endocarditis. A review/removal of all suspect intravascular lines and material is likely indicated at this stage. In cases with initially incomplete source control, we suggest increasing the duration of antibiotics by 5–7 days from the time at which all the sources and septic metastasis were treated and microbiological clearance and clinical improvement were obtained. Some sources require longer antibiotic treatments, such as empyema (4–6 weeks), brain abscesses (6–8 weeks), joint infections including seeding from the BSIs (4–8 weeks) or prosthetic valve endocarditis (4–8 weeks) [2,75].

For some pathogens, extended durations of treatment are warranted. Uncomplicated *S. aureus* BSIs require 2 weeks of antibiotics [76]. Cases with incomplete or ineffective source control or with persisting bacteraemia require 4 and sometimes up to 8 weeks of antibiotics or longer, especially when infected devices or material cannot be removed [2]. For uncomplicated candidaemia, current guidelines recommend 14 days of treatment after the first negative BC [77]. Little data are available for XDR pathogens that have very limited treatment options or that are treated with antibiotics that have lesser activity [72]. For those, it is reasonable to focus on optimal source control and continue treatment for several days after microbiological clearance and clinical improvement.

Severely immunosuppressed patients deserve specific attention. In a cohort study of allogeneic–haematopoietic cell transplant (HCT) recipients with *P. aeruginosa* BSIs or and/or pneumonia, treatment durations of less than 14 days were associated with more recurrent infections [78]. This may not apply to other types of immunosuppression. A cohort study of 249 uncomplicated *P. aeruginosa* BSIs in which 65% of the patients were severely immunosuppressed (3% AIDS, 13% HCT, 21% recent chemotherapy, 16% neutropenia on day 1, 11% other immunosuppressive therapy) did not show any difference in outcomes with shorter compared to longer treatment durations (9 vs. 16 days) [79].

Conversely, for CR-BSIs caused by coagulase-negative staphylococci, a very short treatment of 3 days (or even antibiotic withdrawal) after catheter removal may be sufficient [80,81], highlighting the importance of individualising the treatment duration.

We emphasise that fever and persisting haemodynamic instability after treatment of a BSIs may be also due to an infection at another site or to a non-infectious cause. For all patients who do not show rapid improvement or for those with relapsing sepsis, it is crucial to include those diagnoses in a thorough differential before deciding to continue or escalate antibiotics.

## 6. Conclusions

Blood stream infections in critically ill patients are associated with significant morbidity and mortality. Early adequate antimicrobial therapy, sufficient dosing following ICU specific PK/PD principles and source control are key to improving prognosis. Aggressive ADE and shorter treatments should be used to decrease antibiotic-associated harms.

## Figures and Tables

**Figure 1 antibiotics-11-00362-f001:**
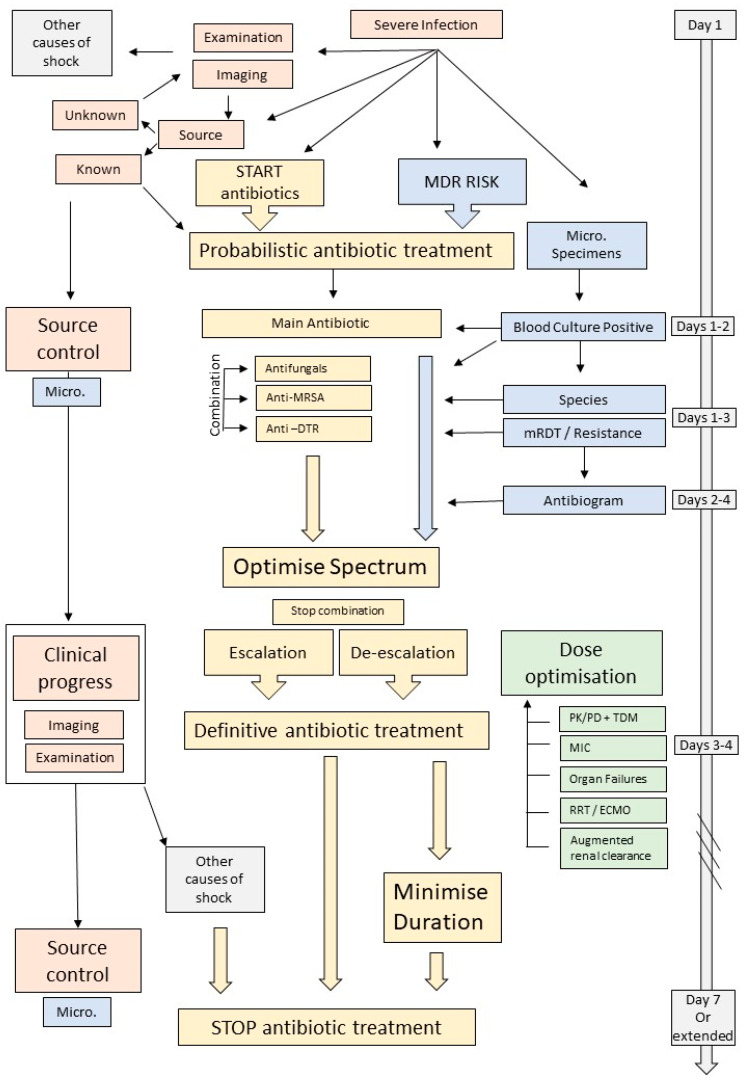
Management of an ICU patient with a blood stream infection. mRDT = molecular rapid diagnostic testing, Micro. = microbiology specimens, MDR = multidrug resistant, DTR = difficult-to-treat resistance, MRSA = methicillin-resistant *Staphylococcus aureus*.

**Figure 2 antibiotics-11-00362-f002:**
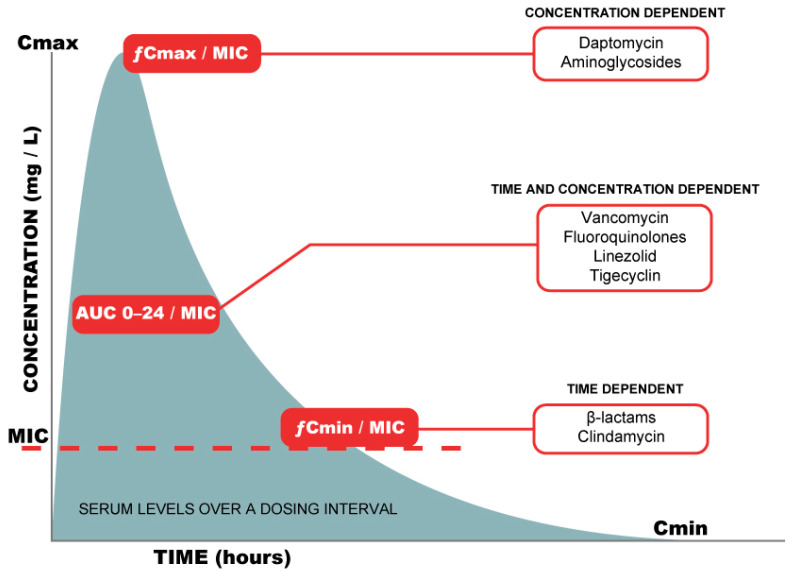
Pharmacokinetic targets for main antibiotic classes. Cmax = maximum serum concentration during a dosing interval, Cmin = trough (minimum) serum concentration over a dosing interval, MIC = minimum inhibitory concentration of the pathogen for the considered antibiotic, fCmax/MIC = ratio of free peak plasma concentration to MIC, fAUC/MIC = ratio of free unbound drug concentration area under the curve to MIC, fT > MIC = free unbound drug concentration time above the MIC.

**Table 1 antibiotics-11-00362-t001:** Most common pathogen groups according to the presumed source of infection.

	Urinary	Respiratory	Intra-Abdominal	Intra Vascular Catheter
Community acquired	Enterobacterales *Enterococcus* sp. *P. aeruginosa* *	*Streptococcus pneumoniae ++* *Legionella* sp. *** Enterobacterales *S. aureus* *P. aeruginosa* * *H. influenzae*	Enterobacterales *Enterococcus* sp. *Candida* sp. Anaerobes Polymicrobial	Coagulase neg. staphylococci *S. aureus* Enterobacterales
Hospital acquired	Enterobacterales *Candida* sp. *Enterococcus* sp. *P. aeruginosa* *Acinetobacter* sp.	Enterobacterales *S. aureus* *P. aeruginosa* *Acinetobacter* sp.	Enterobacterales *P. aeruginosa* *Enterococcus* sp. *Candida* sp. Anaerobes Polymicrobial	Enterobacterales *S. aureus* Coagulase neg. staphylococci *P. aeruginosa* *Acinetobacter* sp.

Describes the most common pathogens. Non-exhaustive list. ++ Largely predominant. * In patients with chronic respiratory disease and patients with long-term indwelling catheter for respiratory and urinary sources, respectively. *** *Legionella* sp. does not cause BSIs but should be included in severe community-acquired respiratory infections.

**Table 2 antibiotics-11-00362-t002:** Risk factors for multidrug-resistant bacteria.

Individual factors (history)	Recent hospitalisation (1 year) Exposure to antimicrobials (3–6 months) Severe co-morbidities (Charlson ≥ 4) Recent immunosuppression Chronic respiratory disease (COPD, cystic fibrosis) Recurrent urinary tract infections Urinary catheter
Individual factors (current)	Prior duration of hospital and ICU stay (continuous increase over time) High severity Known colonisation (surveillance cultures and previous infections)
Institution factors	Regional/institutional prevalence of MDR Overwhelmed health systems

COPD = chronic obstructive pulmonary disease, MDR = multidrug resistant, ICU = intensive care unit.

**Table 3 antibiotics-11-00362-t003:** Targets and dosing strategies for most commonly used antibiotics.

Antimicrobial	Specific Targets	Dosing Strategies	Caution
**Beta-lactam antibiotics**			
Ampicillin–sulbactam	CRAB	9 g q8h (CI/EI)	High dosing increases risk of neurotoxicity
Ampicillin or amoxicillin	Narrow-spectrum targeted therapy	2 g q6h (II)	
Amoxicillin–clavulanic acid	Narrow-spectrum targeted therapy CA-peritonitis	2 g/200 mg q6h (II)	
Piperacillin–tazobactam	Broad-spectrum antipseudomonal probabilistic for HAI	4.5 g q6h EI/CI preferred, loading dose req.	Biliary excretion Resistance promotion
**Antistaphylococcal molecules**			
Flucloxacillin	MSSA	2 g q4–6h (II/CI)	
Cefazolin	MSSA	2 g q8h	
Ceftaroline	MRSA/VISA/VRSE	600 mg q8h	Neutropenia especially in longer treatments
Ceftobiprole	MRSA, MRSE, non-MDR GNB	500 mg q8h (2h EI)	Q4–6 h depending on degree of ARC Dose adjust in renal impairment
Vancomycin	MRSA/MRSE/*E. faecium*	LD 30 mg/kg followed by 30 mg/kg (CI) or 15 mg/kg q12h(II)	TDM required
Daptomycin	MRSA/MRSE/VRE	8–10 mg/kg q24h	
Linezolid	MRSA/MRSE/VRE	600 mg q12h	
**Cephalosporins**			
Ceftriaxone	CAP Susceptible Enterobacterales	1 g q12h EI	
Cefotaxime	CAP Susceptible Enterobacterales	1 g q6h EI CI suggested	
Ceftazidime	*Pseudomonas* sp., *Acinetobacter* sp.	2 g q8h ((EI/CI)	
Cefepime	AmpC-Es	2 g q8h EI	MIC ≥ 4 risk of ESBL-Es and treatment failure Most neurotoxic β-lactam, especially in overdose
Cefiderocol	CREs (KPCs, OXA48, MBLs), DTR-PA	2 g q8h EI (3 h)	Poor efficacy for CRAB
**Carbapenems**			
Imipenem-cilastatin	Broad spectrum Probabilistic for HAI Targeted ESBL-Es *Pseudomonas* sp., *Acinetobacter* sp.	1 g q6–8h (II)	
Meropenem		1–2 g q8h (II, EI, CI)	Poor efficacy against *Enterococcus* sp.
Ertapenem	ESBLE-Es	1–2 g/24 h (II)	
**New combinations** *			
Ceftazidime–avibactam	CREs (KPCs, OXA-48)	2 g/500 mg q8h (II/EI)	
Aztreonam (+CAZ-AVI)	MBL-CREs, DTR-PA, *Stenotrophomonas maltophilia*	2 g q8h	Infuse aztreonam at same time with CAZ-AVI
Ceftolozane–tazobactam	DTR-PA	2 g/1 g q8h (II)	
Aztreonam–avibactam	MBL-CREs	2 g/500 mg q8h (II)	
Meropenem–vaborbactam	KPC-CREs, DTR-PA	2 g/2 g q8h IV (II/EI)	
Imipenem–relebactam	KPC-CREs, DTR-PA	500 mg/250 mg q6h (II)	
**Aminoglycosides**	Combination to extend spectrum when at risk for MDR. ESBL-Es, AmpC-Es, CREs, CRAB, DTR-PA.	Once-Daily dose	Nephrotoxicity Ototoxicity TDM required
Amikacin		25–30 mg/kg (/24h)	
Gentamicin		7–8 mg/kg (/24h)	
**Polymyxins**	CREs (KPCs, OXA48, MBLs) CRAB, DTR-PA Resistant to new/targeted antibiotics		Last-line antimicrobials Nephrotoxicity Use TDM if available
Polymyxin B	Systemic infections	Loading dose 2–2.5 mg/ kg (20,000–25,000 IU/kg) 12-hourly injections of 1.25–1.5 mg/kg (12,500–15,000 IU/kg TBW)	Not renally adjusted Very few data on DTR BSIs
Colistin (CMS)	Urinary source	Loading dose of 300 mg CBA (9 MUI) then 12–24 h later: 300–360 mg CBA/day (9–10 MUI/day) divided in 2 injections	Renally adjusted More nephrotoxicity than polymyxin B
**Other classes**			
Ciprofloxacin	ESBL-Es, AmpC-Es, MDR-PA, *Stenotrophomonas maltophilia*	400 mg q8–12h (II/EI)	
Fosfomycin	CREs (KPCs, OXA48, MBLs) CRAB, DTR-PA		Salvage therapy if susceptible Combination if possible
Tigecycline	CREs (KPCs, OXA48, MBLs) CRAB	100 mg LD then 50 mg q12h OR 200 mg (LD) then 100 mg q12h	Caution with coagulopathy if high dose Use as part of combination
Eravacycline	CREs (KPCs, OXA48, MBLs), CRAB	1 mg/kg q12h (II)	
Cotrimoxazole (TMP/SMX)	ESBL-Es, AmpC-Es, *Stenotrophomonas maltophilia*	1.2–1.6 g SMX q8h (II)	

BSI = blood stream infection, HAI = hospital-acquired infections, CA = community acquired, CAP = community-acquired pneumonia, MDR = multidrug resistant, DTR = difficult-to-treat resistance, MSSA = methicillin-susceptible *Staphylococcus aureus*, MRSA = methicillin-resistant *Staphylococcus aureus*, VISA = vancomycin-intermediate *Staphylococcus aureus*, VRSE = vancomycin-resistant *Staphylococcus aureus*, VRE = vancomycin-resistant *Enterococcus*, PA = *Pseudomonas aeruginosa*, ESBL-Es = ESBL-producing Enterobacterales, CREs = carbapenem-resistant Enterobacterales, CRAB = carbapenem-resistant *Acinetobacter baumannii*, ARC = augmented renal clearance, TDM = therapeutic drug monitoring, LD = loading dose II = intermittent infusion, EI = extended infusion (3 to 4 h), CI = continuous infusion. All EI and CI require a LD, TBW = total body weight, * new refers to recently available BL/BLI combinations targeting specific resistance mechanisms.

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
