# Peer review of "Use of Antimicrobials for Bloodstream Infections in the Intensive Care Unit, a Clinically Oriented Review"

_antibiotics, 2022, doi:10.3390/antibiotics11030362_

Round 1

Reviewer 1 Report

This review conducting a systematic search on Use of antimicrobials for bloodstream infections in the intensive care unit  databases. This review  focusses on the antimicrobial management of BSI whose clinical severity requires ICU admission, or such infections that have been acquired in the ICU. Overall, this manuscript is very interesting and well-written.

Key words: please ICU

Figure 2:  Resolution not good

Author Response

This review conducting a systematic search on Use of antimicrobials for bloodstream infections in the intensive care unit  databases. This review  focusses on the antimicrobial management of BSI whose clinical severity requires ICU admission, or such infections that have been acquired in the ICU. Overall, this manuscript is very interesting and well-written.

Response: we thank the reviewer for these comments.

Key words: please ICU

Response: we have added ICU and Intensive Care to the list of Keywords

Figure 2:  Resolution not good

Response: the figure has been changed to a higher resolution jpeg image.

It is also provided in vectorial format for the editorial office to print in highest possible resolution.

Reviewer 2 Report

First of all, I would like to thank the authors for a very useful paper that will, in my opinion, have an important impact on the antibiotic treatment of patients with BSI on ICU.

The review is well and clearly processed and allows for good orientation in this very important topic.

I recommend accepting the article, but I have a few minor comments and and I consider it appropriate to resolve these.

1) The term species is not written in italics, not even an abbreviation.

2) I consider it appropriate to unify in the text the terms of sp. and spp. Both are possible, but it is not appropriate to mention sp. and a second time spp. At the same time, I recommend mentioning this term in the whole text as well, see line 342 - Acinetobacter species, correctly Acinetobacter sp.

3) If the species name of the bacterium, see Legionella on line 136, is given, the full name must be used, ie Legionella sp.

4) In the Table, I recommend adding a time parameter in the case of "Individual factors and prior duration of hospital and ICU stay", I personally recommend 3-6 months.

5) The term gram-stain should be given in the form Gram stain. Gram is the name of the author of this staining (Hans Christian Gram).

6) In Table 3, I recommend adding to CAP cefotaxime.

7) I consider it appropriate to harmonize the names of Antibiotics (Table 3), for example ampicillin-sulbactam, amoxicillin / clavulanic acid and piperacillin / Tazobactam. Everything should be in a single style. Similarly, polymyxin B should be referred to as Polymyxin B.

8) The correct term is Acinetobacter baumannii, not Acinetobacter Baumannii or Acinetobacter baumanii, line 325.

Author Response

Comments and Suggestions for Authors

First of all, I would like to thank the authors for a very useful paper that will, in my opinion, have an important impact on the antibiotic treatment of patients with BSI on ICU.

The review is well and clearly processed and allows for good orientation in this very important topic.

I recommend accepting the article, but I have a few minor comments and and I consider it appropriate to resolve these.

Response: we thank the reviewer for the careful reading of the manuscript and these comments.

1) The term species is not written in italics, not even an abbreviation.

Response: we thank the reviewer for pointing out this mistake.

All instances (9) of sp.  Have been changed to sp.

2) I consider it appropriate to unify in the text the terms of sp. and spp. Both are possible, but it is not appropriate to mention sp. and a second time spp. At the same time, I recommend mentioning this term in the whole text as well, see line 342 - Acinetobacter species, correctly Acinetobacter sp.

Response: we thank the reviewer for pointing out this mistake.

We have homogenised the text by changing 2 instances of species and 1 instance of spp. to sp.

3) If the species name of the bacterium, see Legionella on line 136, is given, the full name must be used, ie Legionella sp.

Response: thank you, we have added the sp. to the footnotes of the table, where it was missing.

4) In the Table, I recommend adding a time parameter in the case of "Individual factors and prior duration of hospital and ICU stay", I personally recommend 3-6 months.

Response: we thank the reviewer and agree this item required a time bracket to harmonize with the items on the upper part of the table. We have added (Continuous increase over time) rather than (3-6 months).

Indeed, a recent nationwide study from Switzerland of the distribution of pathogens and antimicrobial resistance in ICU-bloodstream infections found that there is a continuous increase in resistance to first line antibiotics from 13.1% on day zero to 60% on days 28-30. This was also found similar results for second line antibiotics.

https://www.nature.com/articles/s41598-021-95873-z

5) The term gram-stain should be given in the form Gram stain. Gram is the name of the author of this staining (Hans Christian Gram).

Response: we thank the reviewer for reminding this important historical fact and honouring the memory of Hans Christian Gram. We have corrected the manuscript accordingly.

6) In Table 3, I recommend adding to CAP cefotaxime.

Response: we have added cefotaxime to table 3 according to the suggestion.

7) I consider it appropriate to harmonize the names of Antibiotics (Table 3), for example ampicillin-sulbactam, amoxicillin / clavulanic acid and piperacillin / Tazobactam. Everything should be in a single style. Similarly, polymyxin B should be referred to as Polymyxin B.

Response: we thank the reviewer for the careful reading of the manuscript. We have corrected those errors and harmonized all antibiotic names using a dash (“-“) with the text and table.

8) The correct term is Acinetobacter baumannii, not Acinetobacter Baumannii or Acinetobacter baumanii, line 325.

Response: we thank the reviewer for pointing this error which has been corrected.

Reviewer 3 Report

The manuscript "Use of antimicrobials for blood stream infections in the intensive care unit" by A. Tabah, J. Lipman , F. Barbier , N. Buetti and J.-F. Timsit is not a review, it's much closer to the expert opinion. There are no any clinical cases, no systematic review and analysis of the literature according PRISMA guidelines. We could not concider this "narrative review" as a review according the journal's rules. Subsequently the work could not be accepted for publication in the current form.

Guess, the work could be re-organized into a Perspectives type according the journal's guideline. The accent on the research-based recomedations and current problems and needs should be added.

The text should be (in my opinion) extensively re-organized too. For example, the last 2 sentences in the Abstract are unnecessary details, it could be deleted.

The main problem is quite simple: we cannot consider the manuscript as a review. Most part of the text is just a classification of different phenomena inside the subject area, but another part includes recommendations and authors' expert opinion. The review in the medical area should be systematic and prepared according the PRISMA guidelines - that's the journal's rules. And the manuscript is absolutely far from this. It could be re-organized as a Perspective - that's another type of papers which allow the expert opinions and opinion-based recommendations inside.

Author Response

Comments and Suggestions for Authors

The manuscript "Use of antimicrobials for blood stream infections in the intensive care unit" by A. Tabah, J. Lipman , F. Barbier , N. Buetti and J.-F. Timsit is not a review, it's much closer to the expert opinion. There are no any clinical cases, no systematic review and analysis of the literature according PRISMA guidelines. We could not concider this "narrative review" as a review according the journal's rules. Subsequently the work could not be accepted for publication in the current form.

Guess, the work could be re-organized into a Perspectives type according the journal's guideline. The accent on the research-based recomedations and current problems and needs should be added.

The text should be (in my opinion) extensively re-organized too. For example, the last 2 sentences in the Abstract are unnecessary details, it could be deleted.

The main problem is quite simple: we cannot consider the manuscript as a review. Most part of the text is just a classification of different phenomena inside the subject area, but another part includes recommendations and authors' expert opinion. The review in the medical area should be systematic and prepared according the PRISMA guidelines - that's the journal's rules. And the manuscript is absolutely far from this. It could be re-organized as a Perspective - that's another type of papers which allow the expert opinions and opinion-based recommendations inside.

Response= We thank the reviewer for the reviewing of our manuscript.

We agree with the reviewer, this is a narrative and not a systematic review. It has been written in a clinically oriented format and should be labelled “clinically oriented narrative review”.

We have added this information

  • To the title
  • To the abstract
  • To the introduction of the main text of the article

We thank the reviewer for the suggestion to re-organize the manuscript and regarding the journal guidelines. We have elected to maintain the structure as is for the two following reasons.

  1. We have checked with the editorial office, who have confirmed “The paper follows the format and structure of our journal”.
  2. A major change with a complete restructure of the manuscript would be at odds with the recommendations of the other reviewers.
  3. The manuscript was written following the instructions to authors as available on the journal webpage and after careful reading of review articles published in the journal to ensure the article would be a good fit for the journal.

We have prepared the manuscript after careful reviewing of the instructions to authors as found on the journal webpage https://www.mdpi.com/journal/antibiotics/instructions .

Reviews: These provide concise and precise updates on the latest progress made in a given area of research. Systematic reviews should follow the PRISMA guidelines. The main text of review papers should be around 4000 words at minimum and include at least two figures or tables.

And

Review manuscripts should comprise the front matter, literature review sections and the back matter. The template file can also be used to prepare the front and back matter of your review manuscript. It is not necessary to follow the remaining structure. Structured reviews and meta-analyses should use the same structure as research articles and ensure they conform to the PRISMA guidelines.

This text implies that non-systematic reviews are acceptable and do require the front and back matter of the template file.

We have prepared the manuscript after careful reviewing of “review” articles that had been published in the journal and found many were in the narrative format. We followed a common format and structure that we found within these manuscripts when preparing the reviewed manuscript by

  1. Describing the issue and setting
  2. Describing how to diagnose it
  3. Describe how to manage it through different sections, providing guidance and describing common pitfalls, specifically discussing the evidence surrounding the use of antibiotics within this population.

A search within the journal was performed with this link:

https://www.mdpi.com/search?journal=antibiotics&article_type=review-article

Examples of narrative reviews outlining a clinical topic of medicine that have been published since December 2021 within the journal are listed below. We believe our article is within a similar format and structure.

  1. Probiotics in the ICU https://www.mdpi.com/2079-6382/11/2/217 (this article is published in the same special issue antimicrobial therapy in the ICU. )
  2. Prevalence and Therapeutic Management of Infections by Multi-Drug-Resistant Organisms (MDROs) in Patients with Liver Cirrhosis: A Narrative Review https://www.mdpi.com/2079-6382/11/2/232/htm
  3. Whole-Person, Urobiome-Centric Therapy for Uncomplicated Urinary Tract Infection https://www.mdpi.com/2079-6382/11/2/218/htm
  4. Treatment of Severe Infections Due to Metallo-Betalactamases Enterobacterales in Critically Ill Patients https://www.mdpi.com/2079-6382/11/2/144/htm
  5. Biofilms in Surgical Site Infections: Recent Advances and Novel Prevention and Eradication Strategies https://www.mdpi.com/2079-6382/11/1/69/htm
  6. Antibiotics and Liver Cirrhosis: What the Physicians Need to Know https://www.mdpi.com/2079-6382/11/1/31/htm
  7. Treating Bacterial Infections with Bacteriophage-Based Enzybiotics: In Vitro, In Vivo and Clinical Application https://www.mdpi.com/2079-6382/10/12/1497/htm
  8. Predicting Antimicrobial Activity at the Target Site: Pharmacokinetic/Pharmacodynamic Indices versus Time–Kill Approaches https://www.mdpi.com/2079-6382/10/12/1485/htm

We thank the reviewer for his time and reviewing of the manuscript. We respectfully hope the reviewer will agree with our response. We remain available for any further questions.

Round 2

Reviewer 3 Report

If the authors and editors suggests that it's possible, I will not against the publication. But I think that it'll be better to mark this work as a Perspective.

I have only minor corrections listed below:

  1. The last 2 sentences in the Abstract are unnecessary details, it could be deleted.
  2. Table 1:  anaerobes and staphylococci are not a taxa names, the itallic does not need;
  3. line 184: sp. . - delete second dot;
  4. line 323: Carbapenem resistant Acinetobacter Baumanii - Carbapenem-resistant Acinetobacter baumanii;
  5. table 3: E. Faecium - E. faecium; Pseudomonas, acinetobacter - Pseudomonas sp., Acinetobacter sp.; S. Maltophilia - S. maltophilia; The same for the legend: please, carefully check lines 445-448;
  6. line 498: coagulase-negative staphylococci, - staphylococci is not a taxa name, the itallic does not need;
  7. lines 525-526: Please double check - delete the sentence.

Author Response

We thank the reviewer for his time, review of our manuscript and attention to detail that have helped improve the quality of the manuscript.

 We provide below a point per point response

  1. The last 2 sentences in the Abstract are unnecessary details, it could be deleted.

The last 2 sentences of the abstract have been removed

  1. Table 1:  anaerobes and staphylococci are not a taxa names, the itallic does not need;

We thank the reviewer and have removed italic for the 2 items anaerobes and staphylococci in table 1

  1. line 184: sp. . - delete second dot;

We have removed the second dot

  1. line 323: Carbapenem resistant Acinetobacter Baumanii - Carbapenem-resistant Acinetobacter baumanii;

We thank the reviewer and have corrected the capitalization and spelling of baumannii.

  1. table 3: E. Faecium - E. faecium; Pseudomonas, acinetobacter - Pseudomonas sp., Acinetobacter sp.; S. Maltophilia - S. maltophilia; The same for the legend: please, carefully check lines 445-448;

We thank the reviewer; multiple capitalisation mistakes and inconsistencies have been corrected

  1. line 498: coagulase-negative staphylococci, - staphylococci is not a taxa name, the itallic does not need;

We have corrected the italic.

  1. lines 525-526: Please double check - delete the sentence.

Both lines and COI have been checked with the author and corrected accordingly.